# Serum Metabolomic Profiling Reveals Differences Between Systemic Sclerosis Patients with Polyneuropathy

**DOI:** 10.3390/ijms26157133

**Published:** 2025-07-24

**Authors:** Kristine Ivanova, Theresa Schiemer, Annija Vaska, Nataļja Kurjāne, Viktorija Kenina, Kristaps Klavins

**Affiliations:** 1Department of Doctoral Studies, Rīga Stradinš University, LV-1050 Rīga, Latvia; 2Institute of Oncology and Molecular Genetics, Rīga Stradinš University, LV-1050 Rīga, Latvia; natalja.kurjane@rsu.lv (N.K.); viktorija.kenina@rsu.lv (V.K.); 3Department of Rheumatology, Pauls Stradiņš Clinical University Hospital, LV-1002 Rīga, Latvia; 4Baltic Biomaterials Centre of Excellence, Headquarters at Riga Technical University, LV-1048 Rīga, Latvia; theresa.schiemer@rtu.lv (T.S.); annija.vaska@rtu.lv (A.V.); kristaps.klavins_3@rtu.lv (K.K.); 5Institute of Biomaterials and Bioengineering, Faculty of Natural Sciences and Technology, Riga Technical University, LV-1048 Rīga, Latvia; 6Department of Biology and Microbiology, Rīga Stradinš University, LV-1050 Rīga, Latvia; 7Centre for Clinical Immunology and Allergy, Pauls Stradiņš Clinical University Hospital, LV-1002 Rīga, Latvia; 8Clinic of Medical Genetics and Prenatal Diagnostics, Children’s Clinical University Hospital, LV-1004 Rīga, Latvia; 9Department of Neurology, Pauls Stradiņš Clinical University Hospital, LV-1002 Rīga, Latvia; 10European Reference Network for Rare Neuromuscular Diseases, 75013 Paris, France

**Keywords:** systemic sclerosis, modified polyneuropathy, peripheral nerve system, metabolome, metabolic profiling, modified Rodnan skin score, Raynaud’s phenomenon

## Abstract

Metabolome studies have already been carried out in patients with systemic sclerosis (SSc). However, polyneuropathy (PNP) as a complication of SSc has been overlooked in these studies. To the best of our knowledge, this is the first study to examine metabolic changes in SSc patients with PNP. Patients with SSc (*n* = 62) and a healthy control group (HC) (*n* = 72) were recruited from two Latvian hospitals. Blood plasma samples were collected and analyzed using an LC-MS-based targeted metabolomics workflow. Our plasma sample cohort consisted of 62 patients with SSc, 42% of whom had PNP. Differences between SSc patients and the HC group with fold changes > 2 were observed for aspartic acid, glutamic acid, valine, and citrulline, all of which were reduced. In contrast to the SSc to HC discrimination, no metabolites had a high fold change; only minor changes were observed using FC > 1.3. We identified elevated concentrations of kynurenine, asparagine, and alanine. Changes in metabolite regulation in patients with SSc, compared to controls, are not identical to those observed in SSc patients with PNP, with elevated concentrations of kynurenine and alanine specific to the SSc subgroup. SSc patients with PNP should probably be considered a distinct population with important metabolomic features.

## 1. Introduction

Systemic sclerosis (SSc) is a rare autoimmune connective tissue disease characterized by vascular insult, autoimmunity, and tissue fibrosis [1,2]. As a complex disorder, it can affect multiple systems, and its clinical presentation varies widely between individuals [1,2]. Several metabolome studies have already been carried out in patients with SSc, revealing changes in several metabolites compared to healthy controls (HCs) [3]. Some of these studies have focused on different manifestations of SSc in patients with interstitial lung disease (ILD) or marked modified Rodnan skin score (mRSS) [4,5]. However, it is notable that polyneuropathy (PNP) as a complication of SSc has been ignored in metabolome studies. PNP, a condition involving multiple peripheral nerves that occurs in SSc, contributes to disability and reduced quality of life [6]. Previous studies have documented the involvement of the nervous system (NS) in SSc, although the prevalence has varied widely, ranging from 17% to 40% [7,8,9,10,11]. One possible reason for this variation is the markedly different approaches to PNP detection, ranging from questionnaires to nerve biopsies [12,13,14,15,16,17,18,19,20]. Therefore, we pre-determined the prevalence of PNP in our study group of SSc patients, which was unexpectedly high [6]. These data further demonstrated the need for metabolome studies in SSc patients with PNP. Understanding the metabolomic alterations in SSc and its related PNP has the potential to uncover novel biomarkers and therapeutic targets, providing opportunities for improved management and outcomes for patients suffering from this complex disease.

Here, to the best of our knowledge, we present the first study of metabolic changes in SSc patients with PNP. Previous studies have included patients with PNP without singling them out in the general SSc population.

The main objective of our study was to determine the metabolome in the SSc patient group compared to HCs and to compare the results with previous metabolome studies in SSc patients. Additionally, our objective was to isolate patients with SSc with diagnosed PNP and to compare the metabolome of this group with patients with SSc without PNP. This approach provided a better understanding of the pathogenesis of PNP in patients with SSc.

## 2. Results

### 2.1. Characteristics of the Study Cohort

Our plasma sample cohort consisted of 62 patients with SSc, of which 26 had PNP (42%). Both total SSc patients and the SSc with PNP subgroup were predominantly females (82% and 77%, respectively); however, PNP was much more common among males. In the SSc with PNP subgroup, the mean age was 15 years higher, and the mean SSc disease duration was 8 years longer than the SSc without PNP subgroup. For further information, see Table 1.

### 2.2. Metabolites in SSc Patients

We first compared plasma metabolites of all SSc patients with HCs, see Table 2.

Based on the PCA analysis (Figure 1a), there was no clear separation between these two groups. However, concentrations of several metabolites changed significantly in the plasma of SSc patients (Figure 1b). The most significant differences with fold changes > 2 were observed for aspartic acid, glutamic acid, valine, and citrulline, all of which were reduced (Figure 1b,c). In particular, the volcano plot showed a general reduction in metabolite concentrations, except glutamine with a fold change >1.5 (A list of all significant metabolites is found in Appendix A).

### 2.3. Disease Prediction Models

Following the hypothesis of using blood plasma metabolites as potential biomarkers for diseases, we tested our dataset on its ability to differentiate SSc from HCs (Appendix A). We used metabolite and two-metabolite ratios to build disease prediction models. We used both significantly changed metabolites (Figure 2a) and metabolites with high predictive scores (Figure 2b) to build models. Model 1 uses metabolites identified through their fold changes, which resulted in the combination of four metabolites: aspartic acid, glutamic acid, glutamine, and carnitine. Model 2 uses metabolites with high predictive scores, combining ornithine and the metabolite ratios glutamine/valine and creatinine/glutamine (Figure 2d). Both models separated patients from controls with an AUC of 0.954 and 0.993, respectively; model 1 gave a slightly better separation of the two groups (Figure 2a,b). The metabolites used for both models did not have an age correlation in SSc, whereas in HCs glutamic acid showed a positive correlation (Appendix A). Despite this, the removal of glutamic acid from model 1 had no impact on model performance (Appendix A).

### 2.4. Discrimination of SSc Patients with PNP

We further subdivided the SSc patients based on the diagnosis of PNP. Again, the PCA analysis shows no separation between subgroups (Figure 3a). In contrast to SSc for the discrimination with HCs, no metabolites had a high fold change (>1.5) or *p*-value (<0.05) used for SSc discrimination. There were minor changes using a lower cutoff of FC > 1.3 and *p*-value < 0.1 [21]. When applying these cutoffs, we identified an elevated concentration of the tryptophan metabolite kynurenine and the amino acids asparagine and alanine (Figure 3b). Kynurenine and alanine were specific for the SSc subgroup with PNP, while asparagine was also found to have a reduced concentration when comparing SSc without PNP with HCs (Figure 3c). These findings prompted us to compare significant changes in the total SSc and SSc subgroup with HCs. Most of the metabolite changes were shared between all groups. Arginine and proline changes were only found in SSc with the PNP subgroup, whereas ornithine was only found in SSc without PNP (Figure 3d). Due to the minor changes in the SSc with the PNP subgroup compared to the SSc without the PNP subgroup, we were unable to construct prediction models that could separate these two groups (Appendix A).

## 3. Discussion

The metabolome, a collection of small compound metabolites in an organism, offers insight into the biochemical changes and potential biomarkers associated with diseases such as SSc [22]. Metabolites can serve as biomarkers for diagnosis, prognosis, and monitoring of disease progression or response to treatment [23]. Analyzing metabolic changes can shed light on the underlying mechanisms of SSc and its complications, including PNP. To our knowledge, this is the first metabolome analysis in SSc patients, with an emphasis on the presence of PNP.

Initially, differences in metabolite regulation were sought between the SSc and HC groups. SSc is a heterogeneous disease with different manifestations and risks of complications. Despite this heterogeneity, previous studies have detected several changes in metabolite regulation in SSc patients. Our study also found several significant differences between SSc patients and HCs. Some of these changes were similar, but some data were distinctly different from previously published data, summarized in Table 3.

We found that the concentration of aspartic acid, or aspartate, was significantly reduced in SSc patients compared to HCs. An important capability of aspartate is to promote macrophage polarization [28]. In SSc, at the peak of the late immune response, endothelin-1 induces polarization of M2, thus potentiating profibrotic activity [29,30]. These results suggest that in SSc, tissue damage is not effectively repaired due to the increased and sustained release of cytokines and growth factors from M2 macrophage cells [31]. In the study by Murgia et al., an analysis of the metabolic profile of SSc patients also showed a reduced concentration of aspartate [5]. The effects of different clinical manifestations of SSc on aspartate levels showed a correlation with mRSS [4]. Significant changes in aspartic acid in patients with SSc detected in our study and in previously published studies may indicate changes in macrophage activation, possibly more pronounced profibrotic activation, as evidenced by correlation with severity of skin involvement, thus signaling macrophage dysregulation.

Another finding in our study was the reduced citrulline concentration in the samples of patients with SSc. Citrulline is an effective substitute for restoring nitric oxide (NO) production in situations of limited arginine availability [32]. NO produced by endothelial cells relaxes vascular smooth muscles, resulting in vasodilation and maintaining the patency of small blood vessels and blood flow through the microvasculature [33]. In SSc, the microvascular bed is the target of an immune-inflammation injury that leads to dysregulation of vascular tone control and results in progressive disorganization of the vascular architecture. In the Smolenska et al. 2019 study, citrulline showed a trend similar to our study, with a lower concentration in patients with scleroderma [25]. In contrast, citrulline was markedly elevated in patients with SSc in the study by Bögl et al. 2022, especially in the diffuse skin SSc group [24]. Although the data from our study may differ from previously published data, the elevated concentration of citrulline may still be associated with the development of skin fibrosis. At the same time, the reduced concentration observed in our study represents an alteration in NO synthesis that could lead to more severe vasculopathy and serve as a marker of vasculopathy in the future.

Carnitine was found to be yet another metabolite with reduced concentration in SSc patients. In the Ottria et al. 2020 study of 27 individuals with SSc, carnitine was elevated in plasma and monocyte-derived dendritic cells [27]. The reduced concentration of carnitine found in our study could be explained by changes in muscle mass in patients with SSc. Not only is the skin and subcutaneous tissue affected, but the normal muscle structure, including smooth muscle and skeletal muscle, is altered, with a general loss of muscle mass [34,35]. Further studies could confirm a correlation between muscle mass and carnitine in patients with SSc.

Valine concentration was reduced in SSc patients. Valine improves cellular mitochondrial function and protects against oxidative stress [36]. There was no significant difference in the Murgia et al. 2018 [5] study between SSc and healthy controls in valine regulation (5). However, patients with diffuse cutaneous SSc (dcSSC) had higher concentrations than patients with limited cutaneous SSC (lcSSc). SSc patients with lung involvement and subclinical pulmonary arterial hypertension (PAH) were found to have higher concentrations of valine as well [4,25].

The last metabolite with reduced concentration in SSc patients was glutamic acid. It is the most abundant central nerve system (CNS) transmitter. Recent data indicate that inflammatory mediators might regulate extracellular glutamic acid concentrations under physiological and pathological conditions [37]. Other studies have also found reduced concentrations of glutamic acid in patients with SSc but higher levels in dcSSc [5,21]. The consensus results of many studies suggest that reduced glutamic acid concentration in SSc patients is not associated with a specific disease complication such as vasculopathy or fibrosis but is a common finding in all SSc patients. It is plausible that these unambiguous changes suggest a role for glutamic acid in SSc immunoregulation and that a reduced concentration of glutamic acid may be one of the markers of persistent damage due to autoimmunity.

Glutamine was the only metabolite with elevated concentration in SSc patients compared to controls. Interestingly, glutamine uptake, but not glutamic acid, is enhanced during T-cell activation [38]. Studying SSc fibroblasts, all showed an increase in glutaminase expression, suggesting that altered glutamine metabolism may be a ubiquitous trait in SSc [39].

Similarly to our study, reduced glutamic acid concentration and elevated glutamine concentration have been reported before [4,5,25]. However, glutamine was one of the few metabolites to have an elevated concentration in patients with lcSSc, compared to patients with dcSSc (5). It is already speculated that elevated glutamine concentration can increase collagen synthesis with subsequent fibrosis of the skin and internal organs [40,41].

In our study, the potential biomarkers identified by fold changes analysis were aspartic acid, glutamic acid, glutamine, and carnitine.

Aspartate has been found to significantly change in SSc patients compared to HCs in other studies as well. Similarly to our findings, aspartate concentration was significantly reduced in SSc patients in the Murgia et al. 2018 study with an AUC > 0.8 [5]. However, Bengtsson et al. found that the concentration of aspartic acid was significantly elevated in SSc patients compared to HCs [26]. This worrying difference could be explained by the small number of SSc patients enrolled (19 subjects) and the significant difference in prior treatment with immunosuppressive agents between studies; in the study by Bengtsson et al., patients had not previously been treated with azathioprine, cyclophosphamide, cyclosporine A, methotrexate, or mycophenolate mofetil [26]. We excluded only patients who previously received cyclophosphamide due to neurotoxicity. We found no similar data on evidence for glutamic acid, glutamine, and carnitine as diagnostic biomarkers in SSc.

We report high predictive scores for the glutamine/valine and creatinine/glutamine ratios. An increased glutamine/valine ratio could indicate increased glutaminolysis, possibly to promote proliferation and altered nitrogen metabolism, as more is studied in cancer studies [42]. We could not find studies with similar data in which two metabolite ratios were used to build disease prediction models. Glutamine was the only metabolite with a significantly elevated concentration in patients with SSc compared to HCs, and by verifying similar data in other studies, we can be more confident of the ability of these metabolite ratios to perform as biomarkers in SSc [5,25].

The findings described above were equivalent to previous metabolome studies in patients with SSc. Our study isolated a previously unstudied group of SSc patients with PNP. There is still no consensus on the pathogenesis of PNP development, so metabolome research may reveal new reasons for the development of neuropathies.

Differences in some metabolites were observed between SSc patients with and without PNP. In contrast to SSc versus HC discrimination, no metabolites had a high fold change (>1.5) or a *p*-value (<0.05). There were minor changes with FC > 1.3 and the *p*-value < 0.1.

Due to the lack of published metabolome studies in SSc that specifically isolate patients with PNP, we first examined altered metabolomes in other metabolome studies of SSc. We summarized the findings in Table 4.

A possible similarity in the development of PNP in patients with SSc lies in the development of diabetic neuropathy (DN). Therefore, we decided to investigate previous metabolome studies in patients with DN, specifically comparing data on altered metabolites in our study in patients with PNP. We summarized the findings in Table 5.

Kynurenine levels were elevated in SSc patients with PNP compared to those without PNP and HCs. The kynurenine pathway, which accounts for the catabolism of approximately 99% of ingested tryptophan not used for protein synthesis, has links with neurodegenerative diseases, tumor proliferation, inflammation, and depression [45]. Possibly due to these findings, the kynurenine pathway is one of the most studied in SSc. Anti-RNA-polymerase III (ARA) positive patients were found to have higher kynurenine levels compared to anti-topoisomerase I and anti-centromere positive patients, as well as SSc patients with dcSSc [46]. Kynurenine levels were higher in PAH patients associated with SSc compared to idiopathic PAH or other connective tissue disease-related PAH and may affect the risk of developing PAH [47,48]. Studies showed that the disturbance of the kynurenine pathway could increase the oxidative compounds, which damage the peripheral nervous system (PNS) and CNS through the broken blood–nerve or blood–brain barrier, respectively [49]. Compared to the effects of the kynurenine pathway in various CNS diseases, data on the role of kynurenine in the development of PNS damage are currently very limited. The concentration of kynurenine was found to be elevated in diabetes mellitus (DM) patients with severe PNP and neuropathic pain [43,44]. The possible elevated concentration of kynurenine also in SSc patients with PNP suggests a unifying dysregulation with PAH, which would be easier to explain due to a common vasculopathy role of both features that are reinforced by kynurenine’s elevated concentration in patients with DN [43,44].

The asparagine concentration was also elevated in patients with PNP compared to SSc patients without PNP, but not to HCs. Asparagine is crucial in proliferating cells when they are starved for nutrients, especially glutamine. Glutamine regulates angiogenesis through multiple mechanisms, and the proliferation of endothelial cells is impaired when exogenous glutamine is unavailable. Instead, endothelial cells rely on asparagine for proliferation, and asparagine can partially rescue these cell defects under low glutamine conditions [50,51]. Unlike other metabolites, asparagine has not been described to have marked changes in SSc and various manifestations of the disease. However, a negative correlation with mRSS was found in SSc patients [4]. In a study with type 2 DM patients, asparagine regulation differentiated between those with and without PNP [44]. It could be inferred that in SSc patients with PNP, an elevated concentration of asparagine signals glutamine deficiency, with changes in endothelial function and regulation of angiogenesis, which could predispose to vasculopathy and ischemic damage as a cornerstone of the development of PNP. However, the elevated glutamine concentration observed in patients with SSc in our study strongly differentiates patients with and without PNP, reinforcing the above hypothesis.

Another metabolite with elevated concentration in SSc patients with PNP, compared to the SSc without the PNP subgroup and HCs, was alanine. Changes in the alanine pathway have been shown to play a role in the development of DN. In a study with type 2 DM patients, the serum β-alanine ratio of β-alanine to L-aspartic acid in DN patients was significantly increased [44]. When present in high levels, β-alanine is a neurotoxin and damages the brain and nerve tissue [52,53,54]. An elevated concentration of alanine, such as observed in patients with type 2 DM, indicates neurotoxic functions of alanine, which may partially explain PNP in SSc or may be a cause of the progression of the disease.

The study’s limitations stem from its strengths. We provided additional data on differences in the metabolome between patients with SSc and HCs. We compared these data with those previously published, thereby strengthening the evidence for the role of metabolome alterations in SSc. However, as the data obtained from the PNP group of SSc patients cannot be compared with similar publications, we chose to investigate the regulation of the altered metabolites found in DN patients. In doing so, we may have overlooked similarities in the development of PNP in patients with SSc that have not yet been published. We acknowledge that the sample size was relatively small, which can be attributed to the rarity and low prevalence of the disease in the Nordic countries. To draw more definitive conclusions about the role of the metabolome in the development and detection of PNP, a larger patient cohort is required. However, PNP in SSc remains under-recognized, underlining the importance of raising awareness of this complication first. Notably, we observed a surprisingly high prevalence of PNP among SSc patients, with age and sex distributions differing from those of SSc patients without PNP.

## 4. Materials and Methods

### 4.1. Subjects

Patients with SSc (*n* = 62) and HC group (*n* = 72) were recruited consecutively at the two leading Latvian hospitals, Pauls Stradiņš Clinical University Hospital and Riga East University Hospital. The study was approved by the Rīga Stradiņš University Research Ethics Committee (Institutional Review Board reference no: 22-2/481/2021), and all participants provided their written informed consent. The diagnosis of SSc was made according to the criteria of the American College of Rheumatology (ACR) and the European League Against Rheumatism (EULAR) [55]. The inclusion criteria for the patients were the diagnosis of SSc according to the criteria of ACR/EULAR and an age of 18 years or older. Most off the patients received treatment with immunosuppressive drugs such as azathioprine, cyclophosphamide, methotrexate, mycophenolate mofetil, and glucocorticoids. Patients with previous treatment with cyclophosphamide, chemotherapy due to cancer, diagnosed DM, thyroid disorders, and stage 4-5 chronic renal disease were excluded. HC inclusion criteria were age 18 years or older without acute infections.

### 4.2. Methods

The enrolled subjects with SSc underwent a uniform PNS evaluation. First, patients underwent a nerve conduction study (NCS) by a certified neurophysiology expert. Motor and sensory conduction were evaluated according to the PNP examination protocol [56]. Each patient underwent an NCS of the bilateral upper extremities (the motor and sensory components of the ulnar and median nerves) and the bilateral lower extremities (the motor component of the peroneal and tibial nerves and the sensory component of the sural nerve) to determine nerve conduction latency, amplitude and velocity. Patients with abnormal results of NCS—considering the normal values used in Latvian clinical practice—in more than one attribute for two separate nerves were diagnosed with PNP. The age at disease onset was defined as the time of onset of the first non-Raynaud’s SSc symptom. A rheumatologist evaluated the skin condition according to the mRSS [57].

### 4.3. Sample Collection and Preparation

Peripheral blood was collected in accordance with the Declaration of Helsinki (1975/83) using an ethylenediamine tetraacetic acid (EDTA) containing BD Vacutainer Blood Collection tube. Plasma separation was performed by centrifuging peripheral blood sample tubes at 4000 rpm, +4 C, for 15 min. Plasma obtained was transferred to −80 °C within 30 min and stored until analysis of the metabolites.

### 4.4. LC-MS Based Metabolomics

For metabolite extraction, 10 μL of plasma samples were mixed with 10 μL of isotopically labeled internal standard mix and 80 μL of methanol. The samples were vortexed for 15 s and centrifuged at 10,000× *g* for 10 min. The supernatant was transferred to HPLC vials and used for LC-MS analysis.

Targeted quantitative metabolite analysis was conducted using HILIC-based liquid chromatography and mass spectrometric detection employing an Orbitrap Exploris 120 system. Metabolites were separated on an ACQUITY UPLC BEH Amide 1.7 μm 2.1 × 100 mm analytical column (Waters, Milford, MA, USA). Gradient elution was carried out using 0.15% formic acid and 10 mM ammonium formate in water as mobile phase A and a solution of 0.15% formic acid and 10 mM ammonium formate in 85% acetonitrile as mobile phase B. The initial conditions were set to 100% mobile phase A. After 6 min, the mobile phase A level was reduced to 94.1%. From 6.1 to 10 min, mobile phase A was set to 82.4%, and from 10 to 12 min, mobile phase A was set to 70.6%. The column was then equilibrated for 6 min at initial conditions. The total analysis time was 18 min. The mobile phase flow rate was 0.4 mL/min; the injection volume was 2 μL, and the column temperature was 40 °C. The MS analysis was performed in ESI positive and ESI negative modes, Full Scan mode with a mass range from 50 to 400 m/z. The ESI spray voltage was set to 3.5 kV in positive mode and 2.5 kV in negative mode, the gas heater temperature was set to 400 °C, the capillary temperature was set to 350 °C, the auxiliary gas flow rate was set to 12 arbitrary units, and nebulizing gas flow rate was set to 50 arbitrary units. For quantitative analysis, seven-point calibration curves with internal standardization were used. Tracefinder 5.1 General Quan (Thermo Fisher Scientific, Waltham, MA, USA) software was used for LC-MS data processing and quantification.

### 4.5. Statistical Analysis

Metabolomics data were analyzed with MetaboAnalyst 6.0 and GraphPad Prism 9.0. Prior to all analysis, metabolites with >50% missing values were removed. For other metabolites, missing values were replaced with 1/5 of the minimal measured value. This imputation technique was chosen based on the assumption that missing values in LC-MS data are typically due to analytes falling under the limit of detection (missing not at random), and such cases seem appropriately handled by determined value replacement [58].

For principal component analysis (PCA) and volcano plots, the data were log_10_ transformed and pareto scaled. Principal components were selected based on parallel analysis, *p*-values and fold changes were plotted as −log_10_ (FC) and log_2_ (*p*), respectively. Metabolite correlation with age was conducted for HCs and SSc patients separately using Pearson’s correlation coefficient (r).

For univariate analysis of SSc to HC, a high significance threshold of FC > 1.5 and *p*-value < 0.05 was chosen. For subgroup analysis, the threshold was lowered to FC > 1.3 and *p*-value < 0.1. This was done to balance capturing minor changes between the groups with restricting false positives as we expect an average of 10% variation for LC-MS data.

For bar plots, original concentrations were normalized to the average concentration of healthy controls for each metabolite. Metabolites were plotted as mean ± SD, and single measurements were overlayed as dots. Significance between groups was determined with 2-way ANOVA, and corrected with Šidká’s multiple comparisons test for two-group comparison, and Tukey’s test for three-group comparisons. Adjusted *p*-values were reported.

For disease prediction models, data were log_10_ transformed and pareto scaled. Models were built using a linear support vector machine (SVM). For exploratory analysis, 6 different models with fixed feature amounts were created and models were averaged from iterations. For curated models, metabolites were selected based either on univariate significance (volcano plots), or average importance scores for disease classification using SVM. The receiver under operating characteristic (ROC) curves and 95% confidence intervals (CI) were calculated from 100-cross validations, and mean ROC curves were reported. The same data were used for training and class prediction visualization.

## 5. Conclusions

Here, we present the first known metabolome study of SSc patients with PNP. This subgroup tended to be older, with a longer duration of SSc, and was more often male.

Analyzing the results of our study and comparing them with previous studies in the field of SSc, we conclude that the metabolite alterations identified in our study are similar to those in other studies, showing associations between macrophage polarization changes, fibrotic process stimulation, and mitochondrial dysfunction with oxidative stress-induced damage.

As potential biomarkers for SSc, we found significant changes in aspartic acid, glutamic acid, glutamine, and carnitine, as well as high predictive scores for ornithine, the glutamine/valine ratio, and the creatinine/glutamine ratio.

Changes in metabolite regulation in patients with SSc, as opposed to controls, are not identical to those observed in patients with SSc with PNP, with kynurenine and alanine displaying elevated concentrations specific to the SSc with PNP subgroup. SSc patients with PNP should probably be considered a distinct population with important metabolomic features. Direct neurotoxicity to PNS structures and mitochondrial dysfunction in conjunction with oxidative stress may play a critical role in the development of PNP in SSc patients according to metabolic profiles, possibly a result of aging and sequential progression of SSc. Nevertheless, further studies are required to evaluate the role of these alterations in the pathophysiology of PNP in patients with SSc to uncover novel biomarkers and therapeutic targets in the future.

## Figures and Tables

**Figure 1 ijms-26-07133-f001:**
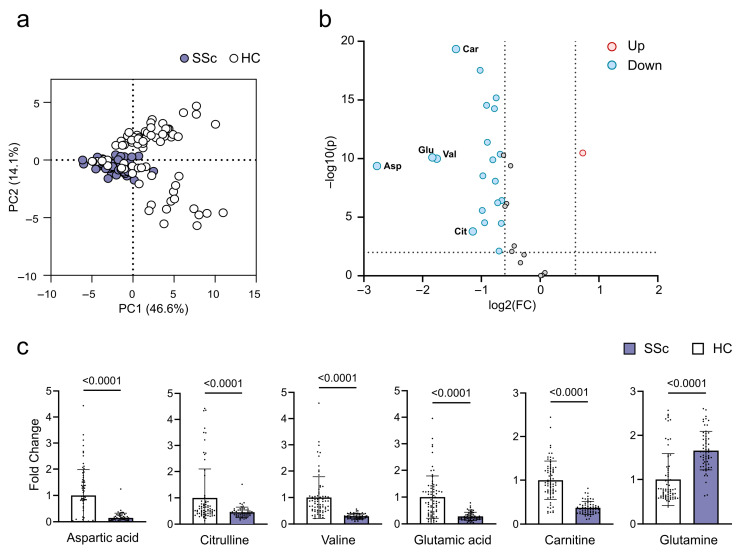
Plasma metabolite changes in systemic sclerosis (SSc) patients compared to healthy controls (HCs). (**a**) PCA plot of SSc patients (purple) and HCs (white), (**b**) Volcano plot of increased (red) and decreased (blue) metabolites using a significance threshold of FC > 1.5 and *p*-value < 0.05; metabolites with FC > 2 are annotated with 3-letter abbreviation. (**c**) Bar plots of metabolites with FC > 2, shown as a fold change relative to the HC average. The bars represent the mean of SSc patients (purple) and HCs (white), individual measurements are overlayed as dots, *p*-values are indicated above each bar. Abbreviations: Asp, aspartic acid; Car, carnitine; Cit, citrulline; Gln, glutamic acid; Val, valine; PC, principal component.

**Figure 2 ijms-26-07133-f002:**
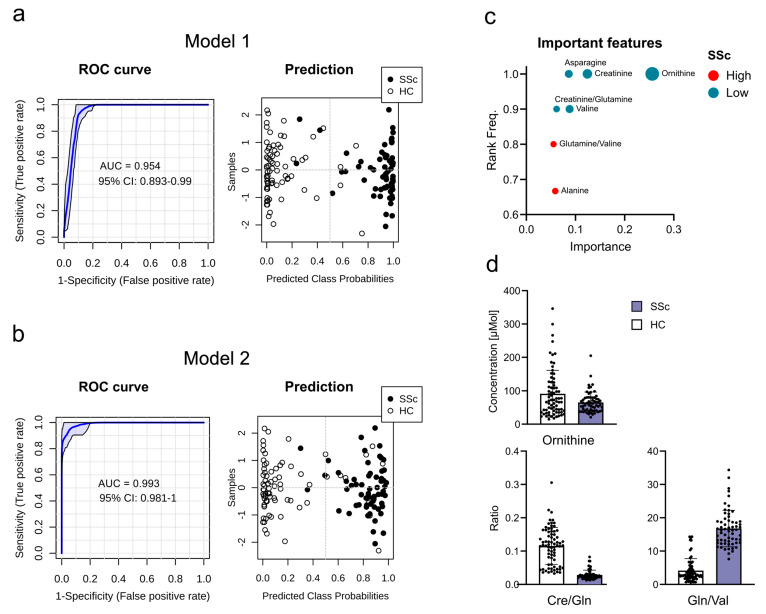
Prediction models distinguishing systemic sclerosis (SSc) patients from healthy controls (HCs). Metabolites were selected based on (**a**) univariant significant changes, or (**b**) high predictive scores. Models were built on linear support vector machine, ROC curves and confidence intervals (CIs) were averaged and calculated from 100 cross-validations, respectively. Class prediction of samples is shown to demonstrate separation power. (**c**) Importance scores of the 7 highest-ranking metabolites for linear support vector machine prediction used for metabolite selection of model 2. (**d**) Plasma metabolite concentrations and their ratios used in model 2. The bars represent mean values for SSc (purple) and HCs (white), individual measurements are overlaid as dots. Abbreviations: Cre, creatinine; Gln, glutamine; Val, valine; ROC, receiver operating characteristic curve; AUC, area under the curve.

**Figure 3 ijms-26-07133-f003:**
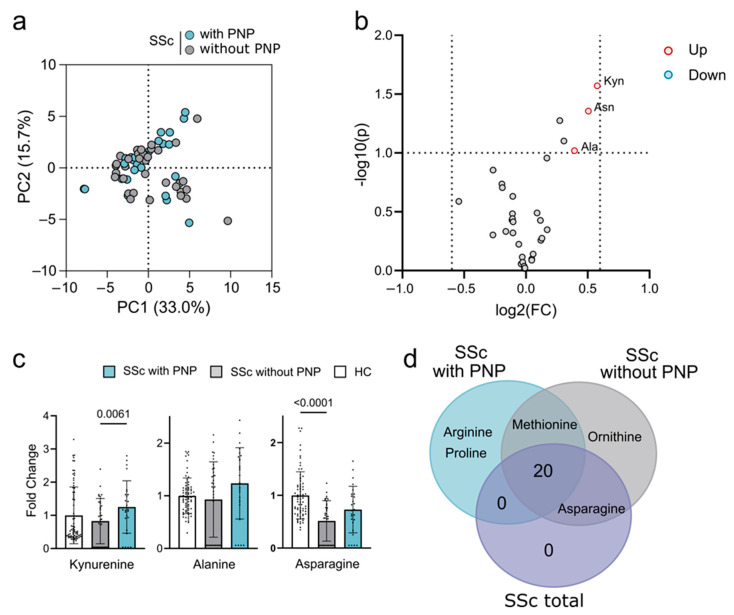
Plasma metabolite changes of SSc patients with and without polyneuropathy (PNP). (**a**) PCA plot of SSc patients with (blue) and without (grey) PNP. (**b**) Volcano plot of increased (red) and decreased (blue) metabolites using a significance threshold of FC > 1.3 and *p*-value < 0.1. Significant metabolites are annotated with a 3-letter abbreviation. (**c**) Bar plots of metabolites with FC > 1.3, shown as a fold change relative to the HC average. The bars represent the mean of SSc patients with (blue) and without (grey) PNP, and HCs (white), individual measurements are overlayed as dots, and *p*-values are indicated above each bar. (**d**) Venn diagram showing the differences from HCs for all SSc patience (purple) and subgroups with (blue) or without (grey) PNP. Abbreviations: Kyn, kynurenine; Asn, asparagine; Ala, alanine.

**Table 1 ijms-26-07133-t001:** Description of the subgroups of the SSc cohort without and with PNP.

Variable	SSc Without PNP	SSc with PNP	*p*-Value
**Sex, n (%)**			0.35
Male	5 (45.5%)	6 (54.5%)	
Female	31 (60.8%)	20 (39.2%)	
**Mean age in years (standard deviation)**	54.94 (12.533)	69.95 (7.893)	<0.05
**Mean disease duration in years (standard deviation)**	10.44 (6.596)	18.08 (10.476)	<0.05
**Mean modified Rodnan skin score (standard deviation)**	7.26 (9.053)	7.65 (8.158)	0.86
**Raynaud’s phenomenon, %**	86	92	0.45

**Table 2 ijms-26-07133-t002:** Description of the SSc cohort group and HC.

Variable	SSc	HC	*p*-Value
**Sex, n (%)**			0.77
Male	11 (17.74%)	14 (19.72%)	
Female	51 (82.26%)	57 (80.28%)	
**Mean age in years (standard deviation)**	61.19 (13.06)	53.18 (18.65)	<0.05

**Table 3 ijms-26-07133-t003:** Comparison of the metabolite changes in SSc found in our study compared to the literature. Metabolites without significant change or not mentioned in the paper are indicated with N.A.

Metabolite	Our Study	Murgia et al. 2018 [5]	Jud et al. 2023 [4]	Guo et al. 2023 [21]	Bögl et al. 2022 [24]	Smolenska et al. 2019 [25]	Bengtsson et al. 2016 [26]	Ottria et al. 2020 [27]
**Aspartic acid or aspartate**	↓	↓	↑ with higher mRSS	N.A.	N.A.	N.A.	↑	N.A.
**Citrulline**	↓	N.A.	correlated with other amino acids	N.A.	↑	↓ with scleroderma	N.A.	N.A.
**Carnitine**	↓	N.A.	N.A.	↑ with higher mRSS	N.A.	N.A.	N.A.	↑
**Valine**	↓	↑ in dsSSc	↑ with a higher DETECT score	N.A.	N.A.	↑ in lung involvement	N.A.	N.A.
**Glutamic acid**	↓	↓ (↑ in dsSSc)	↑ with higher mRSS	↓	N.A.	↑ in calcinosis and telangiectasia	N.A.	N.A.
**Glutamine**	↑	↑	↓ with higher mRSS	N.A.	N.A.	↑	N.A.	N.A.

**Table 4 ijms-26-07133-t004:** Comparison of the metabolites differing between systemic sclerosis (SSc) patients with and without polyneuropathy (PNP) in this study, compared to the literature. Metabolites without significant change or not mentioned in the paper are indicated with N.A.

Metabolite	Our Study	Murgia et al. 2018 [5]	Jud et al. 2023 [4]	Guo et al. 2023 [21]	Bögl et al. 2022 [24]	Smolenska et al. 2019 [25]	Bengtsson et al. 2016 [26]	Ottria et al. 2020 [27]
**Kynurenine**	↑	N.A.	N.A.	N.A.	↑ in dcSSc	N.A.	N.A.	N.A.
**Asparagine**	↑	N.A.	↓ with higher mRSS	N.A.	N.A.	↓ with scleroderma	N.A.	N.A.
**Alanine**	↑	↓	↑ in lcSSc	N.A.	reduced concentration	↑ in dcSSc	↓	N.A.

**Table 5 ijms-26-07133-t005:** Comparison of the metabolites differing between systemic sclerosis (SSc) patients with and without polyneuropathy (PNP) in this study, compared to the metabolite changes in diabetic neuropathy (DN) in the literature. Metabolites without significant change or not mentioned in the paper are indicated with N.A.

Metabolite	Our Study	Staats Pires et al. 2020 [43]	Shao MM et al. 2022 [44]
**Kynurenine**	↑	↑ in DM Type 1 patients with neuropathic pain compared to diabetic controls	↑ in patients with severe DN compared to patients with mild DN; and without DN
**Asparagine**	↑	N.A.	↑ in DM Type 2 patients with PNP compared to DM Type 2 patients without DN
**Alanine**	↑	N.A.	↑ in patients with severe DN compared to patients with mild DN; significantly. ↑ compared to DM Type 2 patients without DN

## Data Availability

The datasets presented in this study are available upon request.

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
