# Peer review of "Serum Metabolomic Profiling Reveals Differences Between Systemic Sclerosis Patients with Polyneuropathy"

_ijms, 2025, doi:10.3390/ijms26157133_

Round 1

Reviewer 1 Report

Comments and Suggestions for Authors

The manuscript by Ivanova et al. presents an analysis of serum metabolomic differences in SSc patients, specifically highlighting polyneuropathy. The data analysis is comprehensive, using established methods appropriately, but there are areas that could be clarified or strengthened for greater rigor and transparency. Following are my specific comments:

Major:

  1. Age and disease duration significantly differ between subgroups (SSc with/without PNP). This could confound metabolomic differences. Have you specifically tested this as covariates in logistic regression models? 

  1. PCA Analysis showed limited separation between groups, but it is unclear whether data were log-transformed or normalized prior to PCA. It would be helpful to define pre-processing steps (log-transformation, normalization, scaling) for PCA analysis.

  1. Thresholds for significance (p-value <0.1 and FC >1.2) are relatively permissive, potentially increasing false-positive findings. Have authors considered performing analysis using alternative methods, such as Wilcox, since FC results are not strong, which may be potentially due to sample size, etc.  I didn't find mention of the FC method used, so it's difficult to comment on this aspect at this point in the review. 

  1. The authors reported disease prediction models with high AUC (>0.93). I could not follow the validation methodology. Could you simplify it or provide more details? To enhance the interpretability of model performance, confusion matrices or sensitivity/specificity data may be provided.

  1. The authors mentioned that missing data were imputed by replacing missing values with 1/5 of the minimal measured value, but justification is lacking. Could you specify why authors don't use more robust imputation methods (e.g., k-nearest neighbors, median/mean substitution, or advanced metabolomics-specific methods)? Maybe performing a sensitivity analysis to confirm that results are robust to different imputation methods.

Minor:

  1. The manuscript figures (PCA and volcano plots) are clear but could benefit from explicit group labeling and clearer figure legends.
  2. Fig 1: The graph's star labelling for P values has 4 stars, and the legend provides a description for 3 stars.

Reviewer 2 Report

Comments and Suggestions for Authors

Thank you for such interesting article. This is the first study to specifically explore metabolomic changes in systemic sclerosis (SSc) patients with polyneuropathy (PNP). The focus on this underexplored subgroup adds genuine novelty to the field. The comparative approach with diabetic neuropathy (DN) studies is a creative strategy to provide context. The study identifies potential biomarkers that could distinguish SSc patients with and without PNP, which may lead to better diagnostic and therapeutic strategies. The findings contribute meaningfully to understanding SSc pathophysiology, especially the neuroinflammatory component.

The paper is clear and well-organized, but I would like to raise some issues for improvement. The manuscript contains minor typographical and formatting issues (e.g., inconsistent line breaks and alignment). The writing could benefit from language polishing for conciseness and clarity.
Moreover, improve table and figure readability:
-- Increase font sizes in figures (especially volcano and ROC plots).
-- Add full metabolite names alongside abbreviations in figure legends for non-expert readers.
-- Table 2 (comparison with prior studies) is highly informative but visually dense. Please consider simplifying the layout or using color coding for increased/decreased concentrations.

According to me, it would be interesting to expand on how these metabolomic findings might translate to clinical practice.
-- Could these metabolites be developed into diagnostic panels?
-- Are there implications for monitoring disease progression or therapy response?
-- What is the biological significance of metabolite ratios (e.g., glutamine/valine)? 

Also, the limitations are acknowledged but could be more detailed in terms of:
-- Sample size, especially for subgroup comparisons (SSc with vs. without PNP);
-- Gender imbalance and age differences between subgroups.

Although multiple comparisons were corrected, consider clarifying whether false discovery rate (FDR) control was used for large-scale metabolite screening.

In addition, justify the fold-change and p-value cutoffs used for identifying significant metabolites, especially the relaxed thresholds for subgroup analysis (PNP vs. non-PNP).

Finally, please consider adding more recent literature on metabolomics-based biomarkers for autoimmune neuropathies if available.

Comments on the Quality of English Language

Thank you for such interesting article. This is the first study to specifically explore metabolomic changes in systemic sclerosis (SSc) patients with polyneuropathy (PNP). The focus on this underexplored subgroup adds genuine novelty to the field. The comparative approach with diabetic neuropathy (DN) studies is a creative strategy to provide context. The study identifies potential biomarkers that could distinguish SSc patients with and without PNP, which may lead to better diagnostic and therapeutic strategies. The findings contribute meaningfully to understanding SSc pathophysiology, especially the neuroinflammatory component.

The paper is clear and well-organized, but I would like to raise some issues for improvement. The manuscript contains minor typographical and formatting issues (e.g., inconsistent line breaks and alignment). The writing could benefit from language polishing for conciseness and clarity.
Moreover, improve table and figure readability:
-- Increase font sizes in figures (especially volcano and ROC plots).
-- Add full metabolite names alongside abbreviations in figure legends for non-expert readers.
-- Table 2 (comparison with prior studies) is highly informative but visually dense. Please consider simplifying the layout or using color coding for increased/decreased concentrations.

According to me, it would be interesting to expand on how these metabolomic findings might translate to clinical practice.
-- Could these metabolites be developed into diagnostic panels?
-- Are there implications for monitoring disease progression or therapy response?
-- What is the biological significance of metabolite ratios (e.g., glutamine/valine)? 

Also, the limitations are acknowledged but could be more detailed in terms of:
-- Sample size, especially for subgroup comparisons (SSc with vs. without PNP);
-- Gender imbalance and age differences between subgroups.

Although multiple comparisons were corrected, consider clarifying whether false discovery rate (FDR) control was used for large-scale metabolite screening.

In addition, justify the fold-change and p-value cutoffs used for identifying significant metabolites, especially the relaxed thresholds for subgroup analysis (PNP vs. non-PNP).

Finally, please consider adding more recent literature on metabolomics-based biomarkers for autoimmune neuropathies if available.

Round 2

Reviewer 1 Report

Comments and Suggestions for Authors

The authors have adequately addressed my previous comments, and I have no further concerns.